# Can education be personalised using pupils' genetic data?

Tim T Morris[1,2]*, Neil M Davies[1,2], George Davey Smith[1,2]

[1]MRC Integrative Epidemiology Unit, University of Bristol, Bristol, United Kingdom; [2]Population Health Sciences, Bristol Medical School, University of Bristol, Bristol, United Kingdom

**Abstract** The increasing predictive power of polygenic scores for education has led to their promotion by some as a potential tool for genetically informed policy. How accurately polygenic scores predict an individual pupil's educational performance conditional on other phenotypic data is however not well understood. Using data from a UK cohort study with data linkage to national schooling records, we investigated how accurately polygenic scores for education predicted pupils' test score achievement. We also assessed the performance of polygenic scores over and above phenotypic data that are available to schools. Across our sample, there was high overlap between the polygenic score and achievement distributions, leading to poor predictive accuracy at the individual level. Prediction of educational outcomes from polygenic scores were inferior to those from parental socioeconomic factors. Conditional on prior achievement, polygenic scores failed to accurately predict later achievement. Our results suggest that while polygenic scores can be informative for identifying group level differences, they currently have limited use for accurately predicting individual educational performance or for personalised education.

## Introduction

The increase in genetic discoveries from large-scale genomewide association studies (GWAS) has greatly advanced scientific understanding of the way in which complex social and health outcomes may arise. GWAS with sample sizes of over one million participants have identified hundreds of genetic variants (single nucleotide polymorphisms, or SNPs) that associate with educational attainment and other social phenotypes (*Lee et al., 2018*; *Karlsson Linnér et al., 2019*; *Luciano et al., 2018*). While individual SNPs associate only very weakly with complex polygenic phenotypes in isolation - typically accounting for less than 0.01% of variation - together they can explain a considerable proportion of phenotypic variation. For example, in the most recent education GWAS, the median per allele effect size of lead SNPs related to an additional 1.7 weeks of schooling, but polygenic scores combining all identified SNPs explained up to 13% of the variance in years of educational attainment (defined as completed years of education) and 9.2% of the variation in educational achievement (defined as high school grade point average [GPA]) in out of sample prediction samples (*Lee et al., 2018*). The combination of multiple SNPs in polygenic scores (*Dudbridge, 2013*) - measures that sum the estimated effects of all individual SNPs associated with a phenotype - are increasingly being used as indicators of genetic propensity, and have been promoted as a potential tool for genetically informed policy (*Plomin, 2018*; *Conley and Fletcher, 2017*). It has been suggested that genetic information could be used prescriptively to provide personalised medicine, education and even dating (*Plomin, 2018*; *McCarthy and Mahajan, 2018*).

Personalisation refers to the tailoring of services away from a one-size-fits-all model to a customised approach that focuses on the needs of an individual. The definition of personalised education has been inconsistent, generally referring to either the tailoring of educational curriculums, learning environments and teaching styles for individual students, or for groups of students within a

*For correspondence:
tim.morris@bristol.ac.uk

**eLife digest** The way that people learn in school and other educational settings differs from person to person for a wide range of reasons. Over the past 15 years there has been a shift in the way that children are taught in the UK and some other countries. Education has become more focused on the students as individuals, recognising that different people learn in different ways and at different speeds. This has led to the idea of 'personalised education', a way of tailoring students' learning to suit their individual needs and differences.

One way that individuals differ is in the genetic material they inherit from their parents. Except for identical twins, no two people have completely identical genomes. It is now easier and cheaper to study the genetic material of individuals than ever before. This has led to a lot of research investigating how our DNA relates to our health, education and other aspects of life.

Some researchers and politicians are now suggesting that individuals' genetic data should be routinely collected by organisations so that their education or health care can be personalised. However, it remains unclear whether this genetic personalisation would be more useful than demographic or socioeconomic data – such as sex, age and family background – that is already available.

To investigate whether an individual's DNA could be used to predict how well they will perform in school, Morris et al. combined genetic data and school test results from a group of 3,500 UK children born in the early 1990s. This revealed that the genetic data did not predict how the children would perform throughout their time at school as accurately as more general information about their family background and other socioeconomic factors.

The findings of Morris et al. suggest that knowledge of students' DNA is unlikely to help educators who want to identify individuals who need extra help or will be at the top of the class. More research is needed on larger groups of children from a broader range of backgrounds, but it is unclear whether a student's DNA will ever be useful for tailoring their education. Currently, it appears that DNA would be less useful for personalising education than easily measured information like test results taken in primary school, education of the child's parents and other social data.

classroom (*Department for Education and Skills, 2004*; *Hartley, 2007*). Throughout, we refer to personalised education as administered at the individual level. Personalised learning was adopted in national policy statements in England in 2004 with a focus on the needs of individual students (*Gilbert, 2020*; *Department for Children, Schools and Families, 2007*). However, it was not mandated and was seen as being conceptually ambiguous, leading to inconsistency in its implementation across schools (*Maguire et al., 2013*). There are currently no policies in place that rely on educational prediction, but calls are increasingly being made for genetic data to be used to personalise education to, for example, identify pupils in need of greater educational support (*Miller, 1990*; *Grigorenko, 2007*; *Sabatello, 2018*). Given the social complexity of educational attainment, polygenic scores for education associate with many aspects of environment and schooling (*Abdellaoui et al., 2019*; *Harden et al., 2020*), referred to as gene-environment correlation. Active gene-environment correlation can be thought of as environment *down*stream of genotype; for example, pupil's selecting certain subjects based on their genotype. Passive gene-environment correlation can be thought of as environment *up*stream of genotype; for example, children of highly educated parents being more likely to inherit education associated environments as well as education associated genes (*Kong et al., 2018*) (also referred to as dynastic effects *Davies et al., 2019*; *Morris et al., 2019*). That a person's education polygenic score associates with a range of phenotypic differences very early in life demonstrates that it captures a very broad range of information, not just their education.

The theoretical maximum bounds placed on the predictive ability of polygenic scores have been discussed in detail elsewhere (see *Janssens et al., 2006*; *Wray et al., 2010*; *Zhao and Zou, 2018*). Briefly, polygenic scores are more predictive when genetic factors play a larger role in a phenotype (as measured by heritability), and in the case of binary phenotypes where prevalence in the outcome is higher (*Janssens et al., 2006*; *Wray et al., 2010*; *Zhao and Zou, 2018*). For polygenic scores to be informative for personalised education and provide actionable information to inform effective

policy, the scores must not only explain sufficient variation in educational achievement across a group of pupils (defined as performance in educational tests), but they must also explain sufficient variation over and above other readily available phenotypic data and accurately predict achievement at the individual level. Phenotypic measures that are predictive of educational achievement such as sex, month of birth and prior achievement (*Benton et al., 2004*; *Solli, 2017*) are readily available to schools, while other measures such as parental education and socioeconomic position (*Morris et al., 2016*; *Strand, 2011*) are, in principle, simple and inexpensive to collect. To date, few studies have investigated how well polygenic scores predict individual level educational attainment or achievement conditional on observable phenotypes that are easily available to educators. Here we investigate how much information pupils' genetics may confer to knowledge of their educational test performance over prior achievement and other phenotypic characteristics.

In this paper we combine educational and genetic data from a UK cohort, the Avon Longitudinal Study of Parents and Children (ALSPAC), to investigate the potential value of genotypic data for predicting pupil achievement and personalising education. We answer three related questions: 1) How predictive of realised educational achievement are polygenic scores? 2) Does polygenic prediction outperform phenotypic prediction from family background measures available to schools? 3) What incremental increase in predictive performance do polygenic scores offer over and above phenotypic information?

## Results

### Group level polygenic score prediction

To investigate how predictive polygenic scores are of realised educational achievement, we created two polygenic scores for education based on the results of the latest GWAS for educational attainment (*Lee et al., 2018*). The first polygenic score used SNPs that reached genomewide significance ($p < 5 \times 10^{-8}$) and the second used all education associated SNPs. Our measure of educational achievement was fine graded point scores from educational exams taken at ages 7 and 16, obtained through data linkage to national school records. The all SNP polygenic score was more strongly correlated with educational achievement ($r$ for age 16 = 0.37) than the genomewide significant polygenic score ($r$ for age 16 = 0.19) (*Table 1*). Children with higher polygenic scores, on average, had higher exam scores than those with lower polygenic scores. Correlations were similar between achievement and parents' years of education and achievement and highest parental socioeconomic position. Correlations were consistently stronger for age 16 than age 7 educational achievement.

Next, we assessed the explanatory power of polygenic scores for educational achievement at age 7. We assessed this using the incremental gain in variance of educational achievement explained by the polygenic scores over and above pupil characteristics available to schools (age, sex, Free School Meal status, English as a Foreign Language status, Special Educational Needs status), parents' years

**Table 1.** Correlation coefficients between educational achievement at ages 7 and 16 and the genotypic and social predictors.

Educational achievement measured using fine graded point scores from educational exams at ages 7 and 16. Genotypic predictors measured using two polygenic scores (PGS) built using only genomewide significant SNPs (GWAS sig PGS) or all education associated SNPs (all SNP PGS) from the largest GWAS of educational attainment (*Lee et al., 2018*). Parental educational attainment (EA) was measured as average completed years of education. Parental socioeconomic position (SEP) was measured as highest parental score on the Cambridge Social Stratification Score scale.

|  | Achievement age 7 | Achievement 16 |
|---|---|---|
| GWAS sig PGS | 0.17 | 0.19 |
| All SNP PGS | 0.26 | 0.37 |
| Mothers EA | 0.28 | 0.39 |
| Fathers EA | 0.27 | 0.40 |
| Parents SEP | 0.30 | 0.40 |

of education, and parent socioeconomic position (*Figure 1*). Both the genomewide significant and the all SNP polygenic scores accounted for a larger proportion of variance explained ($R^2$) in achievement than age and sex alone. Pupil characteristics outperformed polygenic scores in terms of explanatory power, but together they explained up to 21.5% (95% CI: 18.9 to 24.1) of the variation in age 7 achievement. Including information on the social background of pupils' parents that is potentially obtainable by schools further increased the explanatory power of the models up to a maximum $R^2$ of 26.3% (23.4 to 29.2). The incremental $R^2$ of the polygenic scores over pupil characteristics were 1.8% (-0.7 to 4.3) and 4.8% (2.1 to 7.3), suggesting that they provide some additional predictive information over currently available or easily collectable data.

The genomewide significant and all SNP polygenic scores explained more variation of achievement in exams sat at the end of compulsory education at age 16, explaining an additional 3.4% (1.7 to 5.0) and 12.9% (10.6 to 15.3) of educational achievement over age and sex alone (*Figure 2B*). By comparison, measures of parental education and socioeconomic position provided greater returns to explanatory power than the polygenic scores when unadjusted for prior achievement, explaining an additional 19% (16.6 to 21.4) and 21.4% (18.8 to 23.9) respectively over age and sex (*Figure 2B*). As with age 7 achievement, using both genotype and social background data explained the largest amount of variation. At this stage of education schools also hold data on pupils' prior achievement, and these prior achievement measures explained a large amount of variation in age 16 achievement. For example, prior achievement at age 14 explained 65.1% (60.9 to 69.4) of the variation in age 16 achievement alongside age and sex (*Figure 2A*). Conditional on prior achievement data, the polygenic scores provide very little discernible increase in explanatory power (*Figure 2B*).

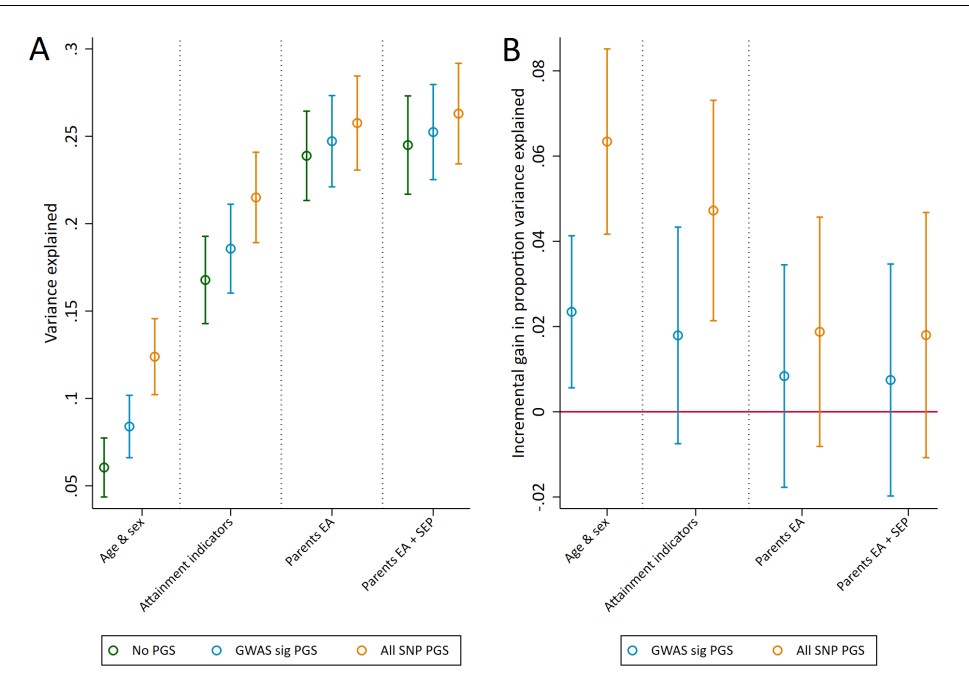

**Figure 1.** Variance in age 7 educational achievement explained by the polygenic scores. (**A**) Variance explained in age 7 educational achievement by the polygenic scores while controlling for pupil characteristics and social factors. (**B**) Additional variance explained by the polygenic scores over and above pupil characteristics and social factors. Educational achievement measured using fine graded point scores from educational exams at age 7. Polygenic scores (PGS) built using only genome-wide significant SNPs (GWAS sig PGS) or all education associated SNPs (all SNP PGS) from the largest GWAS of educational attainment (*Lee et al., 2018*). Pupil characteristics available to schools include Free School Meals (FSM), English as a Foreign language (EFL) and Special Educational Needs (SEN) status. Parental educational attainment was measured as average years of completed education. Parental socioeconomic position (SEP) was measured as highest parental score on the Cambridge Social Stratification Score scale. All analyses include adjustment for the first 20 principal components of population stratification. Parameter estimates in *Supplementary files 1A and 1B*.

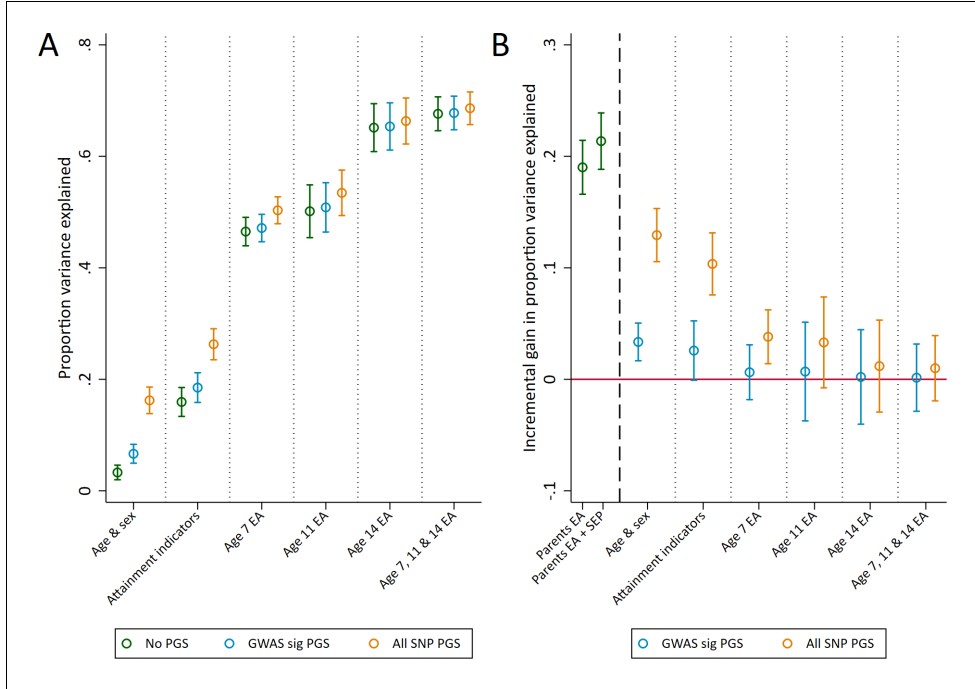

**Figure 2.** Variance in age 16 achievement explained by the polygenic scores. (**A**) Variance explained in age 16 educational achievement by the polygenic scores while controlling for pupil characteristics and social factors. (**B**) Additional variance explained by the polygenic scores over and above pupil characteristics and social factors. Educational achievement (EA) measured using fine graded point scores from educational exams at ages 7, 11, 14 and 16. Polygenic scores (PGS) built using only genome-wide significant SNPs (GWAS sig PGS) or all education associated SNPs (all SNP PGS) from the largest GWAS of educational attainment (*Lee et al., 2018*). Pupil characteristics available to schools include Free School Meals (FSM), English as a Foreign language (EFL) and Special Educational Needs (SEN) status. Parental educational attainment was measured as average years of completed education. Parental socioeconomic position (SEP) was measured as highest parental score on the Cambridge Social Stratification Score scale. All analyses include adjustment for the first 20 principal components of population stratification. Parameter estimates in *Supplementary files 1C and 1D*.

## Individual level polygenic score prediction

We next investigated how well the polygenic scores could identify high achieving pupils, defined as those with the highest 10% of educational test scores. *Figure 3* shows the distributions of the two polygenic scores for high achieving pupils at age 16 and all other pupils. The polygenic scores of high achievers are, on average, higher than of other pupils, but there is near complete overlap in the distributions between the groups. This suggests there would be a large proportion of misclassification when trying to predict from genetic data whether a pupil will be in the top 10%. By comparison, there is far less overlap in the distributions of prior achievement between high achievers and other pupils (*Figure 3—figure supplement 1*). *Figure 4* displays this misclassification of pupils; while some are correctly predicted from their genetic data to be high achievers, a greater proportion are erroneously predicted to be in the wrong group. This misclassification is similar for parental education and socioeconomic position but lower for prior attainment (*Figure 4—figure supplement 1*). In each case, the group of pupils predicted to be in the top 10% of achievers will on average perform higher than other pupils in exams, but the large variability shows that many of the pupils in this group will not ultimately be in the top 10%. High levels of misclassification from the polygenic scores compared to prior attainment were also evident when assessing agreement with quantiled measures of educational achievement (*Supplementary file 1E*).

## Using polygenic scores to identify future pupil performance

To investigate the potential performance of polygenic scores for correctly identifying individual high achieving students from all other pupils, we used Receiver Operating Characteristic (ROC) curves to

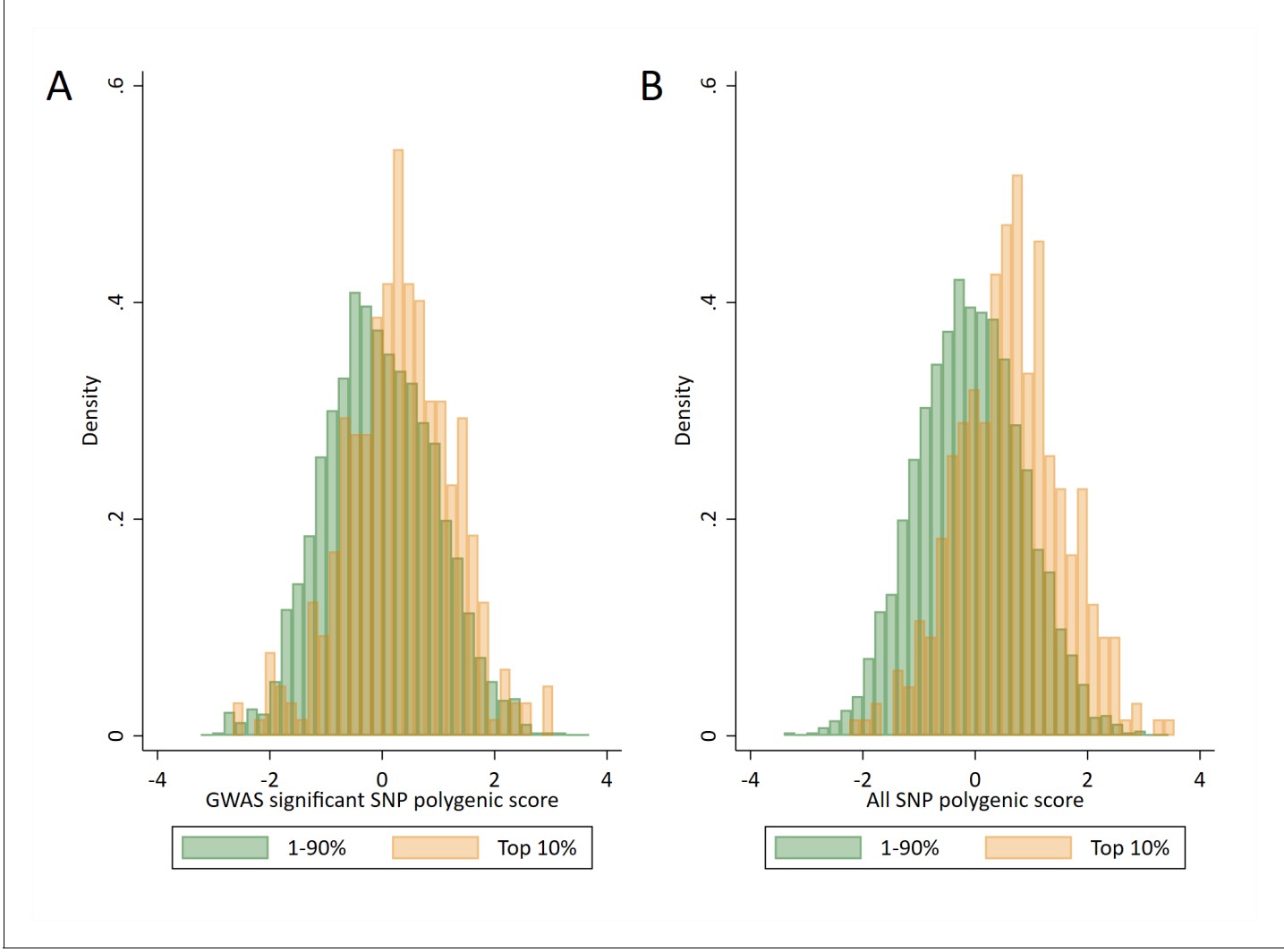

**Figure 3.** Distributions of polygenic scores between 'high achievers' and all other pupils. (**A**) Polygenic score distributions for the GWAS significant polygenic score. (**B**) Polygenic score distributions for the all SNP polygenic score. High achievers defined as pupils with age 16 educational exam scores in the top 10% of the sample. Polygenic scores (PGS) built using only genome-wide significant SNPs (GWAS sig PGS) or all education associated SNPs (all SNP PGS) from the largest GWAS of educational attainment (*Lee et al., 2018*).

The online version of this article includes the following figure supplement(s) for figure 3:

**Figure supplement 1.** Distributions of prior achievement between 'high achievers' at age 16 and all other pupils.

calculate Area Under the Curve (AUC). ROC curves assess the sensitivity (the true positive rate, in our case the probability that a high achieving pupil will be correctly identified as a high achiever) and the specificity (the true negative rate, in our case the probability that that all other pupils will be correctly identified as not being high achievers) of a classifier as its discrimination threshold is varied. Compared to measures of parental socioeconomic position (AUCs: 0.70 for both years of education and social class), the polygenic scores have a lower AUC and therefore poorer sensitivity and specificity to discriminate high achievers at age 7 (AUCs: 0.63 for the GWAS sig PGS; 0.68 for the All SNP PGS) (*Figure 5*). The trade-off in sensitivity and specificity for each of the measures at different classification thresholds is also poor; high sensitivity comes at the cost of low specificity (and vice versa). This means that in order to accurately identify most of the pupils who will go on to be in the top 10% of achievers, one would have to set the classification at the point where almost all students would be identified. These results were consistent when other cut-offs were used to determine the

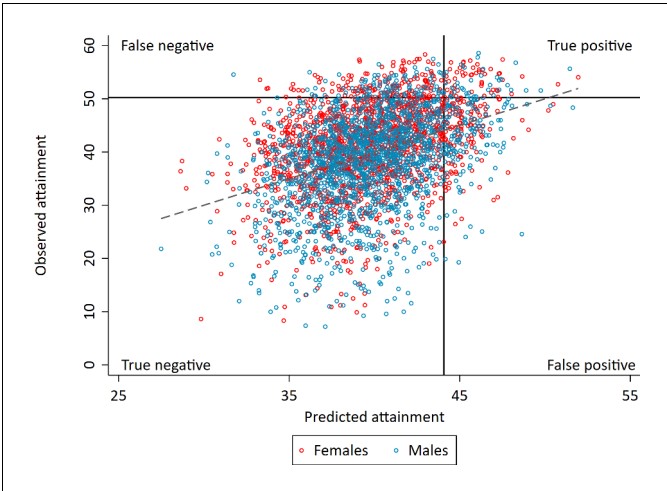

**Figure 4.** Correlation between realised and genetically predicted achievement. Educational achievement measured using fine graded point scores from educational exams at age 16. Predicted achievement at age 16 generated from a polygenic score built using all education associated SNPs (all SNP PGS) from the largest GWAS of educational attainment (*Lee et al., 2018*). Solid lines separate pupils who's exams scores were in the top 10% at age 16 (high achievers) on the y axis and pupils who's exam scores were predicted from genetic data to in the top 10% on the x axis. Dotted line represents best fit.

The online version of this article includes the following figure supplement(s) for figure 4:

**Figure supplement 1.** Independent scatter plots showing correlation between realised achievement at age 16 and achievement predicted from genotypic and phenotypic variables with top 10% of predicted achievers highlighted in green.

high achieving group (*Figure 5—figure supplement 1*), suggesting that the results do not reflect our definition of high achievers.

At age 16, prior achievement data were available. *Figure 6a* shows that measures of prior achievement provide far higher sensitivity and specificity for predicting educational achievement than the polygenic scores (AUCs: 0.83 to 0.95 for prior achievement compared to 0.61 to 0.70 for the polygenic scores). For example, a classification point can be set for prior achievement at age 14 with a sensitivity and specificity of around 0.85, whereas the best classification point for polygenic scores would give a sensitivity and specificity of around 0.65. As with achievement at age 7, the ROC curve for the All SNP polygenic score was similar to the ROC curves for parent's years of education and socioeconomic position (*Figure 6b*). To investigate the value added by polygenic scores above phenotypic data, we calculated ROC curves for the polygenic scores on educational achievement at age 16 residualised on age, sex, prior achievement, and pupil characteristics. The results (*Figure 6c*) demonstrated that after accounting for the phenotypic information already available to schools, the polygenic scores provide almost no information to reliably identify high achievers (AUC: 0.51 and 0.56). The results were consistent had these predictions been made earlier in schooling where later measures of prior attainment were unavailable (*Figure 6—figure supplement 1*; AUC's: 0.54 to 0.61). As with achievement at age 7, these results were consistent when other cut-offs were used to determine the high achieving group (*Figure 6—figure supplement 2*).

If a school headteacher or principal wanted to use polygenic scores as a selection criterion to select the highest performing students, would they identify a group that has higher educational attainment at age 16 than when that selection had been made on other criteria? If they selected the students with the top 10% of polygenic scores, they would on average only sample 24% of the top 10% highest achievers at age 16, and 76% of those not in the top 10%. In contrast, if the principal or policy maker used phenotypic measures from age 11, they would sample 51% of the top 10% highest achievers at age 16, and 49% of those not in the top 10%. This suggests that polygenic scores cannot be used to identify high achieving students more accurately than available phenotypic measures. The group of pupils with the highest polygenic scores do - on average - have higher achievement, but the predictive information provided from the polygenic scores is inferior to that provided

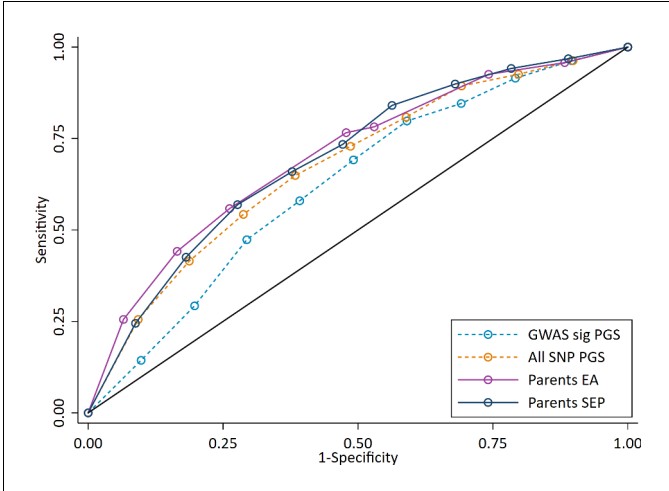

**Figure 5.** ROC curve for being a high achieving student (defined as the top 10% of pupils) at age 7. High achievers defined as pupils with age 16 educational exam scores in the top 10% of the sample. Parental educational attainment (EA) was measured as average years of completed education. Parental socioeconomic position (SEP) was measured as highest parental score on the Cambridge Social Stratification Score scale. Polygenic scores (PGS) built using only genome-wide significant SNPs (GWAS sig PGS) or all education associated SNPs (all SNP PGS) from the largest GWAS of educational attainment (*Lee et al., 2018*). All PGS analyses include adjustment for the first 20 principal components of population stratification. Note that x axis displays 1-specificty. The online version of this article includes the following figure supplement(s) for figure 5:

**Figure supplement 1.** Independent ROC curves for deciled measures of polygenic scores, parental education and parental socioeconomic position predicting high achieving students at age 7 defined at different thresholds (e.g. top 10%).

by phenotypic predictors. *Figure 4—figure supplement 1* demonstrates the variability in age 16 educational achievement for pupils predicted to be in the top 10% from each of the genotypic and phenotypic predictors.

## Discussion

We investigated how predictive polygenic scores for education were of realised educational achievement and the incremental increase in explanatory and predictive power that they offered over and above readily available phenotypic measures. Our results demonstrated that the polygenic scores were predictive of educational achievement, accounting for 3.4% and 12.9% of the variance (above age and sex) across our sample at age 7 and 16 respectively. This was higher than the 9.2% reported for high school GPA in the original GWAS (*Lee et al., 2018*). For informative education predictions at the individual level, the most predictive measure was prior achievement. This reflects some current schooling practices whereby pupils are streamed into different classes based upon ability. Conditional on prior achievement there was little incremental gain in the predictive power of polygenic scores for subsequent achievement, suggesting that when prior achievement data are available, polygenic scores are of little utility to providing accurate predictions of a child's future achievement. When children start school and prior achievement data are unavailable, or in cases where pre-intervention characteristics are limited (*Rietveld et al., 2013*), the scores may provide a small amount of predictive power. However, parental socioeconomic position and education were more strongly predictive of achievement than a pupil's genome. Genetic data from individuals therefore provided little information on their future achievement over phenotypic data that is either available or easily obtainable by educators. This is consistent with results from the only other study we are aware of to assess incremental variance explained over parental social characteristics, which observed higher variance explained in years of education by parental education (18% to 21.3%) than the polygenic score (10.6% to 12.7%) in two US samples (*Lee et al., 2018*).

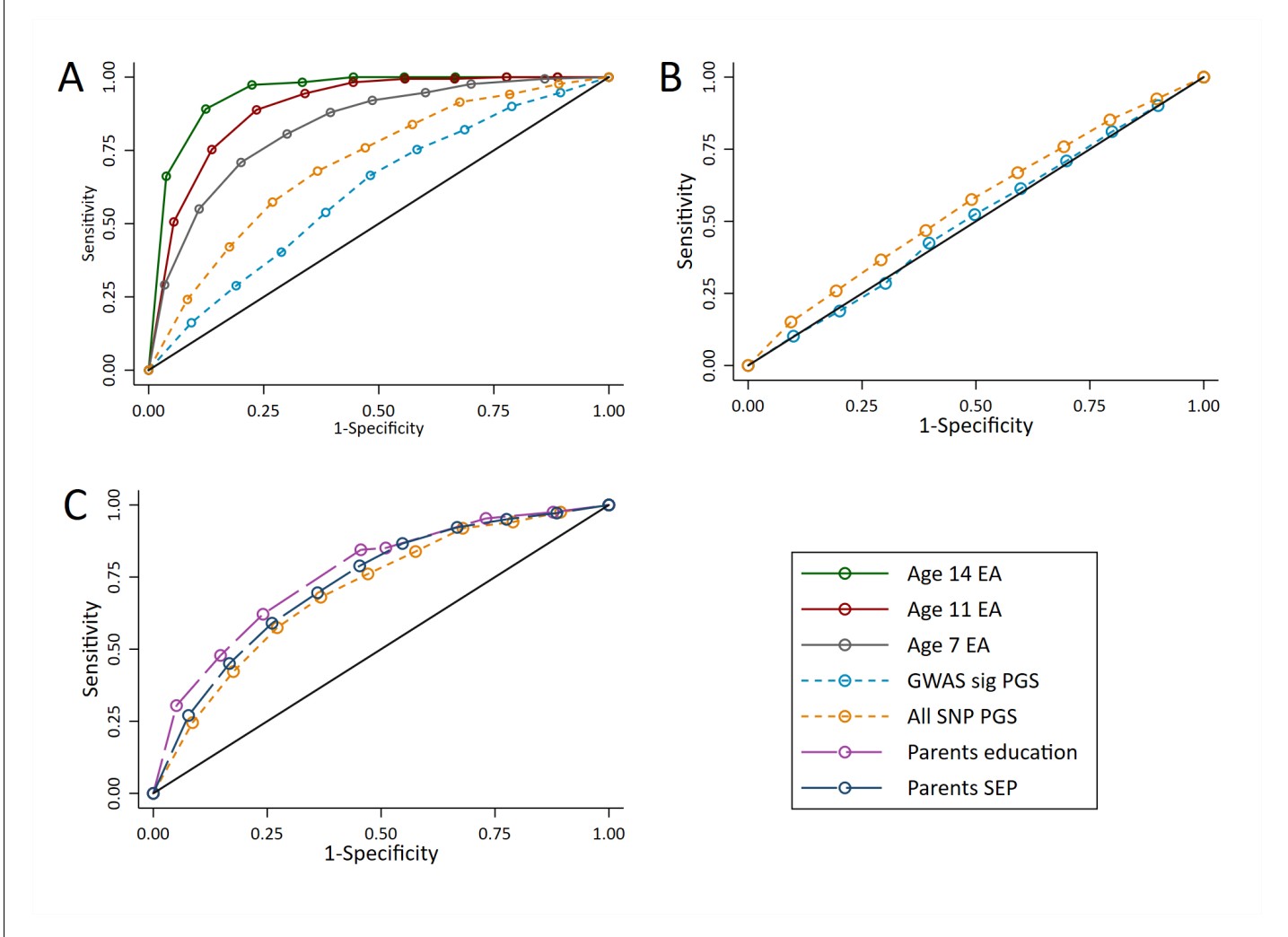

**Figure 6.** ROC curves for being a high achievement student (pupils with age 16 educational exam scores in the top 10% of the sample) at age 16. (**A**) Independent ROC curves for deciled measures of prior achievement and polygenic scores (PGS) predicting high educational achievement (EA) at age 16. (**B**) Independent ROC curves for deciled measures of parental education and socioeconomic position predicting high educational achievement (EA) at age 16. (**C**) ROC curves for deciled polygenic scores predicting high educational achievement (EA) at age 16 residualised on age, sex, prior achievement, and pupil characteristics available to schools. Parental educational attainment was measured as average years of completed education. Parental socioeconomic position (SEP) was measured as highest parental score on the Cambridge Social Stratification Score scale. Polygenic scores (PGS) built using only genome-wide significant SNPs (GWAS sig PGS) or all education associated SNPs (all SNP PGS) from the largest GWAS of educational attainment (*Lee et al., 2018*). All PGS analyses include adjustment for the first 20 principal components of population stratification. Note that x axis displays 1-specificty.

The online version of this article includes the following figure supplement(s) for figure 6:

**Figure supplement 1.** ROC curves for deciled measures of polygenic scores predicting high achieving students at age 16 (pupils with age 16 educational exam scores in the top 10% of the sample) conditional on prior attainment and pupil characteristics.

**Figure supplement 2.** Independent ROC curves for deciled measures of prior attainment and polygenic scores predicting high achieving students at age 16 defined at different thresholds (e.g. top 1%).

The lack of genotypic predictive power that we observed over and above phenotypic data may be because prior achievement mediates genotypic effects on educational outcomes; genetic variants that affect educational achievement at earlier ages are likely to also affect achievement at later ages. It has been suggested that for complex phenotypes, accurate prediction at the individual level may require a polygenic score that explains up to 75% of the total genetic variance of the phenotype (*Wray et al., 2010*). It is therefore possible that polygenic scores for education will require greater

explanatory power for accurate prediction of educational achievement in the future. Our polygenic scores were constructed using results from a GWAS of over a million people, meaning that far larger samples will be required. While future studies may lead to polygenic scores that explain a greater amount of variation in education, these may still not provide useful returns to personalised interventions. High incremental variance explained is a necessary pre-requisite for successful intervention, but it is not a guarantee that an actionable intervention will have a large effect. Furthermore, to provide actionable evidence for personalisation at a given age, polygenic scores need to explain variation in educational outcomes over and above available phenotypic at that age. If most or all the educational differences associated with the polygenic score are phenotypically expressed at a given age, then the score is unlikely to be useful for personalisation.

At the individual level, polygenic scores and parental social background provided similar, but relatively imprecise predictions of achievement within our sample. This reflects a wider issue of the different challenges in analysing group and individual level differences (*Davey Smith, 2011*): while stochastic events will be averaged out at the group level, they are important in determining outcomes at the individual level. There was a large amount of overlap in the polygenic score distribution between pupils in the top 10% of achievers and all others; while pupils with a high polygenic score are more likely to be high achievers, genetics did not determine high achievement. High academic achievement is due to both environmental and genetic factors, including social background (*Morris et al., 2016*), teacher bias (*Campbell, 2015*; *Morris et al., 2018*), the home and school environment (*Nieuwenhuis and Hooimeijer, 2016*; *Rasbash et al., 2010*), and luck (*Davey Smith, 2011*). It is also possible that the quality of family and school environments may constrain or support pupils' ability to exploit their genetic propensity to education. For example, without the means to attend university, it does not matter what an individual's genotype is. In this, it is the combination of nature, nurture and chance that is important (*de Zeeuw and Boomsma, 2017*; *Belsky et al., 2019*).

In fields such as medicine, where genetic risk can be of clinical significance for some diseases (*Lu et al., 2014*), personalisation based on genotype may offer actionable intervention at the individual level. However, our results demonstrate that even for the purpose of identifying groups of pupils who will be high achievers, polygenic scores offer limited prediction value above phenotypic data in education. The usefulness of genetic data for educational research however lies in investigating group level differences. This has been previously demonstrated for example in assessing the effectiveness of teachers and schools (*Harden et al., 2020*; *Morris et al., 2018*); selection differences between schools (*Smith-Woolley et al., 2018*; *Trejo et al., 2018*) social mobility over time and space (*Belsky et al., 2018*), and, in a different context, for performing Mendelian randomization studies of the effects of education on various outcomes (*Tillmann et al., 2017*; *Sanderson et al., 2019*). Our results demonstrate that while polygenic scores are useful for investigating group differences such as these, they do not provide suitable value for routine use by teachers and schools to predict a pupil's future achievement. There is a wide range of non-genetic information available to teachers as part of their day to day interactions with pupils that are used to inform and personalise teaching. This may include knowledge of what the pupil responds well to, any stressful life events that they have recently experienced, and their physical and mental health. To the extent that this knowledge captures genotypic information of the pupil (through its expression in phenotype), it is unclear what novel information genotype would offer to teachers. Finally, genetic studies are focused heavily on samples of European ancestry (*Mills and Rahal, 2019*). Polygenic scores built from these studies do not perform well when applied to other ancestry groups (*Duncan et al., 2019*), meaning that their system-wide application to all pupils in an education system could lead to systematic prediction errors and inequalities in schooling.

This study has several limitations. First, the ALSPAC cohort is not fully representative of the UK population and as such our results may not be generalisable to all UK pupils. Other studies, such as the Millennium Cohort Study are more representative and therefore could provide further evidence about personalised education for the broader UK population. Second, the educational achievement polygenic score that we use was based on a GWAS of years of education rather than exam scores. Years of education can be considered a more social measure of education than exam performance, and previous work has demonstrated that the educational attainment polygenic score strongly reflects parental social position (and through this access to further or higher education) (*Bates et al., 2018*). Future research could investigate this possibility by conducting a GWAS on detailed standardized exam scores on a large sample. Furthermore, it is possible that polygenic scores from a

GWAS conducted on change in test scores throughout education may provide higher prediction accuracy over and beyond phenotypic data if there are genetic factors associated with differences in educational progress. Third, while the educational attainment polygenic score accounts for around 13% of the variance in years of education in our data, increases to this from future GWAS meta-analyses will provide greater power. Twin studies have estimated that the heritability of educational attainment is around 40% (*Branigan et al., 2013*), which limits the predictive power of genetic measures for education over some other phenotypes (*Daetwyler et al., 2008*). Finally, issues from confounding biases caused by population level phenomena such as population stratification, assortative mating and dynastic effects (genetic nurture) (*Kong et al., 2018*; *Morris et al., 2019*; *Bates et al., 2018*; *Young et al., 2018*) may have impacted our results. These biases can lead to social and family differences being masked as genetic differences between individuals, inflating associations between polygenic scores and educational achievement in between individual analyses. Family data are required to further investigate the impact of these baises (*Brumpton et al., 2019*).

In conclusion, our results suggest that currently available genetic scores are unlikely to provide more accurate predictions of how well a pupil will perform in school exams than easily measured phenotypes. Genetic data provide little additional information on an individual's school performance over and above more readily available and easily collected phenotypic data, except where prior achievement measures are unavailable. The greatest value of genetic data may lie instead for researchers investigating the etiology of educational differences between groups of pupils, teachers and schools and for novel sociogenomic analyses into socioeconomic inequalities in education achievement and attainment.

## Materials and methods

### Study sample

Participants were children from the Avon Longitudinal Study of Parents and Children (ALSPAC) (RRID: SCR_007260). Pregnant women resident in Avon, UK with expected dates of delivery 1 st April 1991 to 31st December 1992 were invited to take part in the study. The initial number of pregnancies enrolled was 14,541. When the oldest children were approximately 7 years of age, an attempt was made to bolster the initial sample with eligible cases who had failed to join the study originally. This additional recruitment resulted in a total sample of 15,454 pregnancies, resulting in 14,901 children who were alive at one year of age. From this sample genetic data was available for 7988 after quality control and removal of related individuals. For full details of the cohort profile and study design see *Boyd et al. (2013)* and *Fraser et al. (2013)*. Please note that the study website contains details of all the data that is available through a fully searchable data dictionary and variable search tool at http://www.bristol.ac.uk/alspac/researchers/our-data/. The ALSPAC cohort is largely representative of the UK population when compared with 1991 Census data; there is under representation of some ethnic minorities, single parent families, and those living in rented accommodation (*Boyd et al., 2013*). Ethical approval for the study was obtained from the ALSPAC Ethics and Law Committee and the Local Research Ethics Committees. Following listwise deletion of cases with missing data our final analytical sample was 3,453.

### Genetic data

DNA of the ALSPAC children was extracted from blood, cell line and mouthwash samples, then genotyped using references panels and subjected to standard quality control approaches. ALSPAC children were genotyped using the Illumina HumanHap550 quad chip genotyping platforms by 23andme subcontracting the Wellcome Trust Sanger Institute, Cambridge, UK and the Laboratory Corporation of America, Burlington, NC, US. The resulting raw genome-wide data were subjected to standard quality control methods. Individuals were excluded on the basis of gender mismatches; minimal or excessive heterozygosity; disproportionate levels of individual missingness (>3%) and insufficient sample replication (<0.8). Population stratification was assessed by multidimensional scaling analysis and compared with Hapmap II (release 22) European descent (CEU), Han Chinese, Japanese and Yoruba reference populations; all individuals with non-European ancestry were removed. SNPs with a minor allele frequency of <1%, a call rate of <95% or evidence for violations of Hardy-Weinberg equilibrium ($p<5\times10^{-7}$) were removed. Cryptic relatedness was measured as proportion

of identity by descent (IBD) >0.1. Related subjects that passed all other quality control thresholds were retained during subsequent phasing and imputation. 9115 participants and 500,527 SNPs passed these quality control filters. ALSPAC mothers were genotyped using the Illumina human660W-quad array at Centre National de Génotypage (CNG) and genotypes were called with Illumina GenomeStudio. PLINK (v1.07) was used to carry out quality control measures on an initial set of 10,015 participants and 557,124 directly genotyped SNPs. SNPs were removed if they displayed more than 5% missingness or a Hardy-Weinberg equilibrium P value of less than 1.0e-06. Additionally SNPs with a minor allele frequency of less than 1% were removed. Samples were excluded if they displayed more than 5% missingness, had indeterminate X chromosome heterozygosity or extreme autosomal heterozygosity. Samples showing evidence of population stratification were identified by multidimensional scaling of genome-wide identity by state pairwise distances using the four HapMap populations as a reference, and then excluded. Cryptic relatedness was assessed using an IBD estimate of more than 0.125 which is expected to correspond to roughly 12.5% alleles shared IBD or a relatedness at the first cousin level. Related subjects that passed all other quality control thresholds were retained during subsequent phasing and imputation. 9048 participants and 526,688 SNPs passed these quality control filters.

We combined 477,482 SNP genotypes in common between the sample of mothers and sample of children. We removed SNPs with genotype missingness above 1% due to poor quality (11,396 SNPs removed) and removed a further 321 participants due to potential ID mismatches. This resulted in a dataset of 17,842 participants containing 6305 duos and 465,740 SNPs (112 were removed during liftover and 234 were out of HWE after combination). We estimated haplotypes using ShapeIT (v2.r644) which utilises relatedness during phasing. The phased haplotypes were then imputed to the Haplotype Reference Consortium (HRCr1.1, 2016) panel of approximately 31,000 phased whole genomes. The HRC panel was phased using ShapeIt v2, and the imputation was performed using the Michigan imputation server. This gave 8237 eligible children and 8196 eligible mothers with available genotype data after exclusion of related subjects using cryptic relatedness measures described previously. Principal components were generated by extracting unrelated individuals (IBS <0.05) and independent SNPs with long range LD regions removed, and then calculating using the '–pca' command in plink1.90. Only the children's genetic data was used in this paper.

## Educational achievement

We use average fine graded point scores at four major Key Stages of education in the UK. These are Key Stage 1 (age 7), Key Stage 2 (age 11), Key Stage 3 (age 14), and Key Stage 4 (age 16). We use scores for performance at the end of each Key Stage and a score at entry to Key Stage 1, which represents the start of schooling. At the time the ALSPAC cohort were at school, the age 16 Key Stage 4 exams represented final compulsory schooling examinations. Scores were obtained through data linkage to the UK National Pupil Database (NPD), which represents the most accurate record of individual educational achievement available in the UK. We used data from the Key Stage 1 and Key Stage 4 files. The Key Stage 4 database provides a larger sample size than Key Stage 2 and 3 databases and contains data for each.

## Educational attainment polygenic scores

Two educational attainment polygenic scores were generated using the software package PRSice (*Euesden et al., 2015*) based upon the list of SNPs identified to associate with years of education in the largest GWAS of education to date (*Lee et al., 2018*). The polygenic scores were generated using GWAS results which had removed ALSPAC and 23andMe participants from the meta-analysis (n=763,468), and as such are not perfectly comparable to those reported in the published meta-analysis. SNPs were weighted by their effect size in the replication cohort of the GWAS, and these sizes were summed using allelic scoring. PRSice was used to thin SNPs according to linkage disequilibrium through clumping, where the SNP with the smallest $P$-value in each 250kb window was retained and all other SNPs in linkage disequilibrium with an $r^2$ of >0.1 were removed. The first polygenic score (GWAS sig PGS) was created from the 1,271 independent SNPs that associated with years of education at genome-wide levels of significance ($p<5\times10^{-8}$). The second (all SNP PGS) was created from all genome-wide SNPs reported in the meta-analysis.

## Covariates

We selected covariates that are easily available to schools in the UK. These include the study participants sex and month of birth, and their status on three pupil characteristics that are available to schools the NPD: eligibility for Free School Meals (FSM); Special Education Needs (SEN); and English as a Foreign Language (EFL). FSM is a proxy for low income as only children from low income families are eligible. We use years of parental education, coded as basic formal education (7 years), certificate of secondary education (10 years), O-levels and vocational qualifications (11 years), A-level (13 years), and degree (16 years). For dual parent families we use the average of the two parents' years of education, while for single parent families we use the mother's years of education. Finally, we use a continuous measure of socioeconomic position, the Cambridge Social Stratification Score (CAMSIS). For dual parent families we used the highest of either parents score, while for single parent families we use the mother's score. Parental years of education and CAMSIS were measured when the study participants were in utero.

## Statistical analysis

To examine the predictive ability of polygenic scores for educational achievement we ran a series of regression analyses of the polygenic scores on achievement each controlling for sex, month of birth, and the first 20 principal components of inferred population structure. Principal components are included to adjust estimates for population stratification; systematic differences in allele frequencies between subpopulations due to ancestral differences. Predictive ability of the polygenic scores was determined by the incremental increase in variance explained ($R^2$) in educational achievement above age and sex; pupil characteristics; and prior achievement. Bootstrapping with 1000 replications was used to estimate confidence intervals for $R^2$ values. To compare the predictive power of polygenic scores to additional phenotypic data that schools could collect, we repeated the regression analyses controlling for parental years of education and parental socioeconomic position. Sensitivity and specificity were calculated using selection into the top 10% of educational achievers at age 16 from the whole cohort as the 'diagnosis'. Receiver Operating Characteristic (ROC) curves were used to visually compare models and to calculate the Area Under the Curve (AUC).

## Acknowledgements

We thank Kirsten Leyland for her extremely helpful comments on an earlier version of this manuscript. The Medical Research Council (MRC) and the University of Bristol support the MRC Integrative Epidemiology Unit [MC_UU_12013/1, MC_UU_12013/9, MC_UU_00011/1]. The Economic and Social Research Council (ESRC) support NMD via a Future Research Leaders grant [ES/N000757/1] and TTM via a postdoctoral fellowship [ES/S011021/1]. The Norweign Research Council support NMD via a grant (295989). No funding body has influenced data collection, analysis or its interpretation. This publication is the work of the authors, who serve as the guarantors for the contents of this paper. We are extremely grateful to all the families who took part in this study, the midwives for their help in recruiting them, and the whole ALSPAC team, which includes interviewers, computer and laboratory technicians, clerical workers, research scientists, volunteers, managers, receptionists and nurses. The UK Medical Research Council and Wellcome (Grant ref: 102215/2/13/2) and the University of Bristol provide core support for ALSPAC. A comprehensive list of grants funding is available on the ALSPAC website (http://www.bristol.ac.uk/alspac/external/documents/grant-acknowledgements.pdf). GWAS data was generated by Sample Logistics and Genotyping Facilities at Wellcome Sanger Institute and LabCorp (Laboratory Corporation of America) using support from 23andMe. No funding body has influenced data collection, analysis or its interpretations.

## Additional information

### Competing interests

Neil M Davies: Neil Davies reports a grant for unrelated research from the Global Research Awards for Nicotine Dependence which is an Independent Competitive Grants Program supported by Pfizer. The other authors declare that no competing interests exist.

## Funding

| Funder | Grant reference number | Author |
|---|---|---|
| Economic and Social Research Council | ES/S011021/1 | Tim T Morris |
| Medical Research Council | MC_UU_12013/1 | Tim T Morris<br>Neil Martin Davies<br>George Davey Smith |
| Wellcome | MC_UU_12013/1 | Tim T Morris<br>Neil Martin Davies<br>George Davey Smith |
| Medical Research Council | MC_UU_12013/9 | Tim T Morris<br>Neil Martin Davies<br>George Davey Smith |
| Wellcome | MC_UU_12013/9 | Tim T Morris<br>Neil Martin Davies<br>George Davey Smith |
| Medical Research Council | MC_UU_00011/1 | Tim T Morris<br>Neil Martin Davies<br>George Davey Smith |
| Wellcome | MC_UU_00011/1 | Tim T Morris<br>Neil Martin Davies<br>George Davey Smith |
| Economic and Social Research Council | ES/N000757/1 | Neil Martin Davies |
| Norwegian Research Council | 295989 | Neil Martin Davies |

The funders had no role in study design, data collection and interpretation, or the decision to submit the work for publication.

## Author contributions

Tim T Morris, Conceptualization, Data curation, Formal analysis, Investigation, Visualization, Methodology; Neil M Davies, George Davey Smith, Conceptualization, Supervision, Funding acquisition, Investigation, Methodology

## Author ORCIDs

Tim T Morris (iD) https://orcid.org/0000-0001-8178-6815
Neil M Davies (iD) https://orcid.org/0000-0002-2460-0508
George Davey Smith (iD) https://orcid.org/0000-0002-1407-8314

## Ethics

Human subjects: Informed consent was obtained from all participants. Ethical approval for the study was obtained from the ALSPAC Ethics and Law Committee and the Local Research Ethics Committees. Relevant Research Ethics Committee approval references are as follows: Bristol and Weston Health Authority: E1808 Children of the Nineties: Avon Longitudinal Study of Pregnancy and Childhood (ALSPAC); Southmead Health Authority: 49/89 Children of the Nineties - "ALSPAC"; Frenchay Health Authority: 90/8 Children of the Nineties; North West 5 Research Ethics Committee: 10/H1010/70 (protocol Number 1278); Project to Enhance ALSPAC through Record Linkage (PEARL): phenotypic enrichment of the ALSPAC Cohort though linkage to primary care electronic patient records and other databases.

## Decision letter and Author response

Decision letter https://doi.org/10.7554/eLife.49962.sa1
Author response https://doi.org/10.7554/eLife.49962.sa2

## Additional files

### Supplementary files

• Supplementary file 1. Full parameter estimates for all models presented throughout the manuscript. (**A**) Variance explained in educational achievement at age 7. Polygenic scores (PGS) built using only genome-wide significant SNPs (GWAS sig PGS) or all education associated SNPs (All SNP PGS) from the largest GWAS of educational attainment (*Lee et al., 2018*). FSM, Free School Meals. EFL, English as a Foreign Language. SEN, Special Educational Needs. Eduyears, years of education. SEP, socioeconomic position. (**B**) Incremental R2 for educational achievement at age 7. Polygenic scores (PGS) built using only genome-wide significant SNPs (GWAS sig PGS) or all education associated SNPs (All SNP PGS) from the largest GWAS of educational attainment (*Lee et al., 2018*). FSM, Free School Meals. EFL, English as a Foreign Language. SEN, Special Educational Needs. Eduyears, years of education. SEP, socioeconomic position. (**C**) Variance explained in educational achievement at age 16. Polygenic scores (PGS) built using only genome-wide significant SNPs (GWAS sig PGS) or all education associated SNPs (All SNP PGS) from the largest GWAS of educational attainment (*Lee et al., 2018*). FSM, Free School Meals. EFL, English as a Foreign Language. SEN, Special Educational Needs. EA7, prior educational achievement at age 7. EA11, prior educational achievement at age 11. EA14, prior educational achievement at age 14. Eduyears, years of education. SEP, socioeconomic position. (**D**) Incremental R2 for educational achievement at age 16. Polygenic scores (PGS) built using only genome-wide significant SNPs (GWAS sig PGS) or all education associated SNPs (All SNP PGS) from the largest GWAS of educational attainment (*Lee et al., 2018*). FSM, Free School Meals. EFL, English as a Foreign Language. SEN, Special Educational Needs. EA7, prior educational achievement at age 7. EA11, prior educational achievement at age 11. EA14, prior educational achievement at age 14. (**E**) Agreement between educational achievement (EA) quantiles and other quantiled measures. Polygenic scores (PGS) built using only genome-wide significant SNPs (GWAS sig PGS) or all education associated SNPs (All SNP PGS) from the largest GWAS of educational attainment (*Lee et al., 2018*). p values for all tests < 0.001.

• Transparent reporting form

### Data availability

We are unable to make any of the the data used in the study publicly available due to data sharing restrictions imposed by the conditions of access to ALSPAC data. The ALSPAC data used for this research project is however available to all bonafide researchers subject to access approval of a study proposal by the ALSPAC Executive committee (alspac-exec@bristol.ac.uk). For full details of the application process to obtain ALSPAC data please see the ALSPAC access policy available on the study website at .

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

## Appendix 1

### Supplementary material

We split educational achievement and the polygenic scores by quintiles and deciles to assess the level of agreement (*Supplementary file 1*). The agreement between the quantiled measures of achievement at ages 7 and 16 and the polygenic scores were slightly higher than expected. For the GWAS significant polygenic score, the Kappa statistics show that agreement was at most only 5% higher than would be expected by random agreement compared to perfect agreement (quintiles $\kappa$ = 0.05 for achievement at age 7). Agreement was higher for the all SNP polygenic score and generally higher for age 16 than age 7 achievement. At age 16, agreement was at least twice as high for quantiles of prior achievement than the polygenic scores when compared to random allocation. Agreement with age 16 achievement was highest for age 14 achievement, being 46% and 28% better than expected by chance for quintiles and deciles of achievement respectively.

