## [Decision Letter]

Thank you for submitting your article "Can education be personalised using pupils' genetic data?" for consideration by *eLife*. Your article has been reviewed by three peer reviewers, and the evaluation has been overseen by a Reviewing Editor and Mark McCarthy as the Senior Editor. The following individuals involved in review of your submission have agreed to reveal their identity: Cecile Janssens (Reviewer #1); Kathryn Paige Harden (Reviewer #2); David Cesarini (Reviewer #3).

The reviewers have discussed the reviews with one another and the Reviewing Editor has drafted this decision to help you prepare a revised submission.

Summary:

All reviewers agree that this is an interesting manuscript with clear research questions and solid methods. The paper sets out the limits of genetic prediction for aspects of educational attainment, especially as compared to the existing information supplied by readily available risk predictors (and previous educational performance).

Having said that the reviewers considered that the manuscript needed to be revised to reflect the issues described below. The thrust of the argument from several of the reviewers (both in their reviews and in the subsequent inter-reviewer discussion) was a "shared concern that the paper is a bit cavalier in how it talks about the relationship between incremental R^2^s and the scope for personalized interventions. The title is an obvious place to start fixing this issue". One reviewer suggested a more descriptively on-point title along the lines of: "Polygenic Scores For Educational Attainment Do Not Predict Exam Performance Beyond Readily Available Phenotypic Information". All felt that the discussion about personalized interventions could use a major overhaul that takes into account the specific concerns raised.

Essential revisions:

The major issues that need to be addressed are these:

1) More complete analysis of group level predictions (see more detailed comments from R1 in particular). The key point here is that in focusing on individual level predictions (which are "far out of reach" as anticipated given the low R2), an opportunity has been lost to explore the impact on group level differences (and how the PRS might, or might not, influence that above and beyond standard criteria). As reviewer 1 states: "by focusing on individual prediction (which is way out of reach), we lose out of sight that there is potential for misuse on group level discrimination"

2) Defend the use of cutoffs such as defining the outcome as the top 10%? Does the ROC change with different definitions of the outcome?

3) Please define "personalized education". Per the comments of reviewer 2, you should spell out what you mean by this term (is it individual personalization within each classroom, or is it streaming of groups of students), and give reference to the educational literature on this subject. This revision is clearly tied to the first, in terms of setting out more clearly the opportunities for use/misuse of PRS in both the individual and the group setting. As reviewer 2 states: "the authors could be more careful about delineating more specifically *which* educational policies or practices hinge on individual prediction."

4) Please do more to help readers understand exactly what is being reported in a few instances. The goal is a paper clear enough that someone with access to the data set could reproduce the exact analyses run without having to make guesses or judgment calls. (see full reviews for more specific suggestions).

5) Please do more to help readers compare their estimates to the previous literature.

6) Please expand the paper's Discussion about the relationship between the incremental R2 of a PGS and the scope for personalized interventions (see full reviews for details).

A suggestion from reviewer 1 with which we agree:

7) The structure of the results can be improved by presenting three sections in line with the three research questions. The ROC analyses is discussed under the individual prediction, but it is a population evaluation equivalent to (but not the same as) R2, not for individual prediction.

---

## [Author Response]

Essential revisions:The major issues that need to be addressed are these:1) More complete analysis of group level predictions (see more detailed comments from R1 in particular). The key point here is that in focusing on individual level predictions (which are "far out of reach" as anticipated given the low R2), an opportunity has been lost to explore the impact on group level differences (and how the PRS might, or might not, influence that above and beyond standard criteria). As reviewer 1 states: "by focusing on individual prediction (which is way out of reach), we lose out of sight that there is potential for misuse on group level discrimination"

We have conducted additional analyses that are presented in the Materials and methods and are referenced in the Results section. Our key point is that conditional on observable phenotypes after the age of 11, the polygenic scores do not appear to be informative. Figure 6—figure supplement 1 demonstrates that even for group level analysis, at age 7 (the first Key Stage of schooling in the UK) one could not reliably differentiate future high achievers from other students better than by selecting on phenotypic achievement. In the Discussion we explain that education is already personalised to a certain extent by teachers, but that this personalisation occurs on phenotypic characteristics such as observed attainment/aptitude and attitude to school. Specifically, we have made changes to the Results section with the following passages of text:

“This misclassification is similar for parental education and socioeconomic position but lower for prior attainment (Figure 4—figure supplement 1). In each case, as a group the pupils predicted to be in the top 10% of achievers will on average perform higher than other pupils in exams, but the large variability shows that many of the pupils in this group will underperform.”

“If a school headteacher or principal wanted to use polygenic scores as a selection criterion to select the highest performing students, would they identify a group that has higher educational attainment at age 16 than when that selection had been made on other criteria? If they selected the students with the top 10% of polygenic scores, they would on average only sample 24% of the top 10% highest achievers at age 16, and 76% of those not in the top 10%. In contrast, if the principal or policy maker used phenotypic measures from age 11, they would sample 51% of the top 10% highest achievers at age 16, and 49% of those not in the top 10%. This suggests that polygenic scores cannot be used to identify high achieving students more accurately than available phenotypic measures. The group of pupils with the highest polygenic scores do – on average – have higher achievement, but the predictive information provided from the polygenic scores is inferior to that provided by phenotypic predictors. Figure 4—figure supplement 1 demonstrates the variability in age 16 educational achievement for pupils predicted to be in the top 10% from each of the genotypic and phenotypic predictors”.

2) Defend the use of cutoffs such as defining the outcome as the top 10%? Does the ROC change with different definitions of the outcome?

To assess whether polygenic scores could be used to correctly identify a pupil as being in an actionable group (whether that be predicted low attainer in need of support or a predicted high attainer to be streamed for advanced classes) we require a cut-off for this group. This requirement for a binary outcome for use in the ROC analysis is what motivated our use of cut-offs. Given that our choice of the top 10% may seem arbitrary we have now repeated the analyses at a range of cut-offs (1%, 5%, 10%, 25%, 50%, 75%, 90%, 95%, 99%). These are presented in the Materials and methods (Figure 5—figure supplement 1 and Figure 6—figure supplement 1) and show that our results are consistent across the group definitions. These results are presented in the Materials and methods and discussed in the main body of text, with the following inclusions:

“These results were consistent when other cut-offs were used to determine the high attaining group (Figure 5—figure supplement 1), suggesting that the results do not reflect our definition of high-attainers.”

“As with achievement at age 7, these results were consistent when other cut-offs were used to determine the high attaining group (Figure 6—figure supplement 2).”

3) Please define "personalized education". Per the comments of reviewer 2, you should spell out what you mean by this term (is it individual personalization within each classroom, or is it streaming of groups of students), and give reference to the educational literature on this subject. This revision is clearly tied to the first, in terms of setting out more clearly the opportunities for use/misuse of PRS in both the individual and the group setting. As reviewer 2 states: "the authors could be more careful about delineating more specifically *which* educational policies or practices hinge on individual prediction."

We have now expanded upon “personalised education” and have clearly defined this in the Introduction, as follows:

“The definition of personalised education has been inconsistent, generally referring to either the tailoring of educational curriculums, learning environments and teaching styles for individual students, or for groups of students within a classroom. Personalised learning was adopted in national policy statements in England in 2004 with a focus on the needs of individual students. However, it was not mandated and was seen as being conceptually ambiguous, leading to inconsistency in its implementation across schools. Throughout, we refer to personalised education as administered at the individual level. There are currently no policies in place that rely on educational prediction, but calls are increasingly being made for genetic data to be used to personalise education.”

We appreciate reviewer #2’s comments that potential policy implications could in theory be based on the average PGS of a classroom rather than the PGS of an individual (in line with the classroom focussed definition of personalised education). We have chosen not to discuss this in the manuscript as we are unable to think of a scenario where a class average PGS could provide actionable information. Children are streamed into classes based on performance, which on this evidence will provide more useful and accurate information of future attainment than a PGS.

4) Please do more to help readers understand exactly what is being reported in a few instances. The goal is a paper clear enough that someone with access to the data set could reproduce the exact analyses run without having to make guesses or judgment calls. (see full reviews for more specific suggestions).

We have edited the manuscript throughout to improve the narrative and make our analyses clearer to the reader. We have also now referred to our measure of educational performance as achievement rather than attainment, and we have made clearer what terms such as “incremental R2” refer to.

5) Please do more to help readers compare their estimates to the previous literature.

We have tied our results into the previous literature as best we can, now referring to the prediction in the Lee et al., 2018, GWAS as it refers to high school GPA which we feel makes a better comparison than years of education. We have also drawn comparisons to the Lee et al., 2018, incremental R2 analysis which reviewer #3 helpfully pointed us towards. We feel it pertinent to note that while personalised education has been discussed previously in papers, other than the analysis performed by Lee et al., 2018, we are aware of little for us to compare our results to.

In response to reviewer #3, the figure of 13.8% was erroneous and has now been corrected.

6) Please expand the paper's Discussion about the relationship between the incremental R2 of a PGS and the scope for personalized interventions (see full reviews for details).

We have now expanded upon this issue in our Discussion. We have added the following text:

“While future studies may lead to polygenic scores that explain a greater amount of variation in education, these may still not provide useful returns to personalised interventions. High incremental variance explained is a necessary pre-requisite for successful intervention, but it is not a guarantee that an actionable intervention will have a large effect. Furthermore, to provide actionable evidence for personalisation at a given age, polygenic scores need to explain variation in educational outcomes over and above available phenotypic at that age. If most or all the educational differences associated with the polygenic score are phenotypically expressed at a given age, then the score is unlikely to be useful for personalisation”.

A suggestion from reviewer 1 with which we agree:7) The structure of the results can be improved by presenting three sections in line with the three research questions. The ROC analyses is discussed under the individual prediction, but it is a population evaluation equivalent to (but not the same as) R2, not for individual prediction.

Thanks for this suggestion, we have now restructured the results as suggested with a third section titled “Using polygenic scores to identify future pupil performance”.